# Regioselective Synthesis of Potential Non-Quinonoid Prodrugs of Plasmodione: Antiparasitic Properties Against Two Hemoglobin-Feeding Parasites and Drug Metabolism Studies

**DOI:** 10.3390/molecules29225268

**Published:** 2024-11-07

**Authors:** Elena Cesar-Rodo, Baptiste Dupouy, Cécile Häberli, Jean-Marc Strub, David L. Williams, Pascal Mäser, Matthias Rottmann, Jennifer Keiser, Don Antoine Lanfranchi, Elisabeth Davioud-Charvet

**Affiliations:** 1Laboratoire d’Innovation Moléculaire et Applications (LIMA), Team Bio(IN)organic & Medicinal Chemistry, UMR7042 CNRS-Université de Strasbourg-Université Haute-Alsace, European School of Chemistry, Polymers and Materials (ECPM), 25, Rue Becquerel, F-67087 Strasbourg, France; 2Swiss Tropical and Public Health Institute, Kreuzstrasse 2, CH-4123 Allschwil, Switzerlandpascal.maeser@swisstph.ch (P.M.); matthias.rottmann@swisstph.ch (M.R.); jennifer.keiser@swisstph.ch (J.K.); 3Laboratoire de Spectrométrie de Masse BioOrganique (LSMBO), IPHC UMR 7178 CNRS, Université de Strasbourg, F-67087 Strasbourg, France; 4Department of Microbial Pathogens and Immunity, Rush University Medical Center, 1735 West Harrison Street, Chicago, IL 60612, USA; david_williams@rush.edu; 5University of Basel, Petersgraben 1, CH-4001 Basel, Switzerland

**Keywords:** angular methyl, antiplasmodial, antischistosomal, Diels–Alder cycloaddition, hemoglobin, plasmodione, prodrug

## Abstract

Ψ-1,4-naphthoquinones (Ψ-NQ) are non-quinoid compounds in which aromaticity—found in 1,4-naphthoquinones—is broken by the introduction of an angular methyl at C-4a or -8a. This series was designed to act as prodrugs of 1,4-naphthoquinones in an oxidative environment. Furthermore, from a medicinal chemistry point of view, the loss of planarity of the scaffold might lead to an improved solubility and circumvent the bad reputation of quinones in the pharmaceutical industry. In this work, we illustrated the concept by the synthesis of Ψ -plasmodione regioisomers as prodrugs of the antimalarial plasmodione. The presence of a chiral center introduces a new degree of freedom to be controlled by enantioselectivity and regioselectivity of the cycloaddition in the Diels–Alder reaction. The first strategy that was followed was based on the use of a chiral enantiopure sulfoxide to govern the stereoselective formation of (+)Ψ-NQ or (−)Ψ-NQ, depending on the chirality of the sulfoxide (*R* or *S*). New sulfinylquinones were synthesized but were found to be ineffective in undergoing cycloaddition with different dienes under a wide range of conditions (thermal, Lewis acid). The second strategy was based on the use of boronic acid-substituted benzoquinones as auxiliaries to control the regioselectivity. Using this methodology to prepare the (±)Ψ-NQ racemates, promising results (very fast cycloaddition time: ~2 h) were obtained with boronic acid-based quinones **25** and **27** in the presence of 1-methoxy-1,3-butadiene, to generate the 4a- and the 8a-Ψ-plasmodione regioisomers **1** and **2** (synthesized in six steps with a total yield of 10.5% and 4.1%, respectively. As the expected prodrug effect can only be revealed if the molecule undergoes an oxidation of the angular methyl, e.g., in blood-feeding parasites that digest hemoglobin from the host, the antimalarial and the antischistosomal properties of both (±)Ψ-NQ regioisomers were determined in drug assays with *Plasmodium falciparum* and *Schistosoma mansoni*. Metabolic studies under quasi-physiological conditions and LC-MS analyses were undertaken to reveal the generation of plasmodione from both the 4a- and the 8a-Ψ-plasmodione regioisomers.

## 1. Introduction

Malaria is a human parasitic disease caused by the protozoan *Plasmodium* spp. parasites, which are transmitted through the bites of infected mosquitoes. It is mostly found in sub-Saharan Africa, as well as tropical and sub-tropical areas, and there were an estimated 249 million cases of malaria that occurred in 85 malaria-endemic countries in 2022 [1]. An estimated 608,000 deaths occurred globally due to malaria in 2022, a mortality rate that mainly affects infants under five years of age in sub-Saharan Africa. Five major species of the *Plasmodium* parasite can infect humans; the most serious forms of the disease—cerebral malaria, severe anemia and death—are largely caused by *P. falciparum*. Most of the drugs available to treat malaria are subject to resistance, which limits their effectiveness [2]. In the last decade, our team has invested considerable effort in a series of potent compounds known as 3-benzylmenadiones, targeting the redox equilibrium in *P. falciparum*-infecting red blood cells. Most of our research has focused on the lead molecule plasmodione (PD) [3] (Figure 1) by developing chemical methodologies [4,5] to optimize new pharmacophores and by understanding its mode(s) of action [6,7]. Another neglected disease, schistosomiasis (also called bilharziasis), is responsible for 251.4 million people million people infected in many tropical areas, resulting in 11,792 deaths globally per year [8]. However, these figures are likely underestimates of the current status of the disease and need to be reassessed. Schistosomiasis is due to five main species of flatworms (platyhelminth parasites), causing intestinal and urogenital infections in humans: *Schistosoma mansoni*, *S. japonicum*, *S. haematobium*, *S. intercalatum* and *S. mekongi*. The intermediate hosts of this hemoglobin-feeding parasite are of various species of freshwater snails, which transfer human infection through skin contact with infective cercariae [9]. Praziquantel has been used to treat schistosomiasis since the 1980s, but this drug is not effective against immature worms and is currently the only available medication, jeopardizing the treatment if parasite drug resistance increases worldwide.

Plasmodione (PD) is a redox-active prodrug (Figure 1) [5,7]. Its bioactivation in parasitized red blood cells involves a cascade of redox reactions generating a 3-benzoylmenadione (or plasmodione oxide, PDO) that interacts with hemoglobin digestion. Both pathways, i.e., drug bioactivation and hemoglobin catabolism, were demonstrated to produce a large amount of reactive oxygen species (ROS) in the parasites [7]. The drug-induced oxidative stress was visualized in the apicoplast and the cytosol of the parasites using GFP-based biosensors [6,7]. Nevertheless, the high specificity of PD is based on the quasi-absence of methemoglobin in non-parasitized red blood cells, in accordance with its very low toxicity in glucose-6-phosphate dehydrogenase-sufficient or -deficient red blood cells [7].

Plasmodione contains both a quinone moiety, working as an electron reservoir for redox cycling, and an aromatic planar structure, which contributes to poor aqueous solubility and high lipophilicity. The electrophilic and oxidant properties of unsubstituted quinones are well documented: they contribute to instability and high reactivity in cells due to prior enzyme-catalyzed reduction of the quinone, causing multiple issues and a directed toxicity [10]. Their ability to undergo multiple cycles of reduction and oxidation can damage and induce oxidative stress around the molecule before reaching their envisaged biological targets. To balance this dilemma, many electrochemical studies have shown that the electron-acceptor properties of quinones, causing the formation of radical semiquinone anion or dihydroquinone dianion species responsible for in vivo oxidative stress, can be modulated by electron-withdrawing or -donating substituents of the electroactive core [5]. As previously observed for most of the 3-benzylmenadiones, plasmodione is not highly oxidant, in accordance with the absence of initial liability of toxicity in mammal cells [3]. After bioactivation through reduction and benzylic oxidation, redox-cycling generates a 3-benzoylmenadione (plasmodione oxide, PDO) with higher oxidant character. The electrochemical data recorded for diverse 3-acylated menadiones attest to a marked impact of the 4′-benzoyl substitution on the second electron transfer, leading to the dihydronaphthoquinone dianion (to up to 500 mV) [5].

In this work, we introduce an angular methyl to break the aromatic core of the quinone moiety of menadione (pseudo-menadione or Ψ-menadione). We illustrate our exploration with the synthesis of both non-quinonoid plasmodione regioisomers, called here pseudo-plasmodiones **1** and **2** (4a- or 8a- Ψ-plasmodiones, respectively) (Figure 1). The introduction of an angular methyl can be beneficial for physicochemical and pharmacokinetic properties (PK). Indeed, by breaking the aromaticity of a molecule, its planarity and its three-dimensional geometry are directly affected, which could lead to an increase of the aqueous solubility of the molecule [11]. Furthermore, the angular methyl can be seen as a shield for direct bioreduction or nucleophilic attack, providing a stable molecule until its bioactivation in an oxidative environment, such as that found in parasites. With the aim to build a prodrug of plasmodione (PD), we postulated that the angular methyl can be hydroxylated in situ by an aromatase-like activity present in the parasite, possibly catalyzed by hemoglobin in large excess, and therefore generate the active plasmodione in situ in the parasite (Figure 1). This known catalytic process, e.g., in the formation of estrogens after aromatization of androgens, is catalyzed by aromatase [12] belonging to the cytochrome P-450 family [13]. The catalytic introduction of a double bond through oxidation of a methyl group followed by a C-C cleavage was also observed in fungal ergosterol synthesis involving the sterol 14α-demethylases (CYP51) [14]. To make this hydroxylation possible, CYP450 contains an iron of the prosthetic group, i.e., the molecule of heme in the Fe^III^ form, then reductase (NADPH) occurs, where dioxygen works in pair with iron in a redox chain reaction to form a ferryl Fe^III^-OOH and make the acyl-carbon cleavage possible. At the clinical level, women with breast cancer, treated by aromatase inhibitors, were observed to express increased hemoglobin levels, likely to compensate the inhibition of aromatase [15,16].

In the present work, we describe the synthesis of two regioisomeric analogues of plasmodione, the 4a-Ψ-plasmodione **1** and the 8a-Ψ-plasmodione **2** (Figure 1). Furthermore, both the 4a-Ψ-plasmodione **1** and the 8a-Ψ-plasmodione **2** were tested for antimalarial and antischistosomal activities against blood-feeding parasites, *P. falciparum* and *S. mansoni*, two parasites that digest hemoglobin from the host and produce large amounts of hemozoin. We postulated that the angular methyl of the Ψ-plasmodiones could be metabolized by an aromatase-like protein present in the parasite producing the active plasmodione through a beneficial prodrug effect, and thus, we carried out biomimetic oxidation reactions in the presence of the 4a- (or 8a-) Ψ-plasmodione **1** (or **2**), hematin, the NADPH-dependent *P. falciparum* Ferredoxin-NADP^+^ Reductase (*Pf*FNR)-catalyzed PDO system (Figure 1) [6].

## 2. Results and Discussion

From a chemical view, the real challenge is the introduction of a chiral center and, thereby, the control of the stereochemical and regiochemical course during the synthetic route. The structure of 1,4-naphthoquinone is a well-documented recurrent building block in [4 + 2] Diels–Alder (DA) cycloadditions, and controlling its regio/stereo selectivity is an essential task already described in the literature [4,17,18,19,20]. We drew on two different methodologies, both originally described by Carreňo and co-workers [21,22]. The authors and some of us [23] were able to obtain an angular methyl-bearing enantiopure benzoquinone with a high enantiomeric excess from a Diels–Alder cycloaddition mediated by a sulfinylquinone (Figure 2, Route A).

This methodology, known as the introduction of chiral sulfinyl onto the dienophile, induces a regio/stereoselectivity of the reaction, which has been applied in various applications [24,25]. Carreňo and co-workers also described the regioselective production of Ψ-naphthoquinones (Ψ-NQ) with an angular methyl group from boronic acids [20,22]. In this reaction, the boronic acid-substituted dienophile is more reactive than the sulfinylquinone to control the regioselectivity course of the reaction and self-elimination by protodeborylation, but product **3** is obtained as a racemate (Figure 2, Route B).

Therefore, we investigated these two methodologies (Figure 3), depicted in Routes A and B, and established a successful pathway for the synthesis of the 4a- and 8a-Ψ-plasmodione regioisomers **1** and **2** (Routes C and D).

### 2.1. Preliminary Investigations: Synthesis of the Ψ-Menadiones

First, we tried to obtain the Ψ-menadiones using the methodology described by Carreňo’s group [20,22] from the corresponding benzoquinone-substituted boronic acids **5** and **6** (Figure 4). Boronic acid **5** was synthesized from the corresponding commercially available benzoquinone precursor, i.e., the 2,5-dimethylbenzoquinone, by bromination with Br_2_ leading to **7**. Then, after a two-step reduction/protection, the dimethoxy-bromobenzene **8** was engaged through a Br/Li exchange protocol in the reaction with B(O*i*Pr)_3_, followed by acidic hydrolysis of the boronated ester. Subsequent oxidative demethylation of **9** with cerium ammonium nitrate (CAN) gave the desired product **5**, which was isolated after four steps in a 30% overall yield (Figure 4). Concerning the other regioisomer **6**, another synthetic route was performed [20,22], starting from the corresponding benzoquinone with a reduction of the quinone moiety followed by a bromination with *N*-bromosuccinimide (NBS) leading to **10**. Subsequent protection of the hydroxyl group gave **11** and finally the same two steps-pathway, as used for **5**, was applied to obtain **6** after four steps in 18% overall yield (Figure 4).

Both freshly prepared benzoquinone-substituted boronic acids were engaged in the Diels–Alder cycloaddition with 1-methoxy-1,3-butadiene and, after a short reaction time, led to compounds **3** and **13** in good to excellent yields (Figure 5). Carreňo’s group described a domino process including the Diels–Alder cycloaddition reaction, a protodeboronation and an elimination of MeOH [20,22]. This exclusive reactivity under mild conditions in a short period of time without any Lewis acid as catalyst is explained by the activation of the dienophile through a hydrogen bond between the boronic acid and quinoid carbonyl, which could not be established with a boron-free species [26]. Then, the regioselectivity of the reaction is fully controlled by the boron, with the *endo* attack of the diene from both faces (TS, Figure 5). For example, the bottom attack could produce an undetected *ortho-cis* intermediate (A and B, Figure 5) with the benzoquinone substituent facing upwards. From the reported study [20,22], the reaction had been performed in an NMR tube in CD_2_Cl_2_, to demonstrate that the protodeboronated adduct was formed directly in the crude reaction before any workup. From different X-ray structures, this protodeboronation took place in *trans* compared to the boron. Intermediate C (Figure 5) *meta-tran*s was then proposed, with a subsequent *beta*-*cis* elimination of MeOH producing the desired product. Since the diene can attack from the top face of the dienophile, the two enantiomers were obtained, therefore **3** and **13** are racemates.

### 2.2. Synthesis of Ψ-Plasmodione Regioisomers from Sulfoxide-Substituted 3-Benzyl-benzoquinones

As drawn in Figure 3 (route A) and detailed in Figure 6, we first functionalized the starting dienophile by the Kochi–Anderson benzylation [4] before the Diels–Alder cycloaddition. Carreňo’s group [21] described the regio- and enantio-selective cycloaddition to access an enantiopure anthraquinone from chiral sulfinylquinone. Lanfranchi and Hanquet supported these selectivity results [23,27], which were due to the specific orientation of the chiral sulfinyl moiety, blocking one face and the other producing a single enantiomer. In the reported methodology, an *endo* attack was described producing an *ortho-cis* intermediate with an attack of the diene from the less hindered face, which is the top face, leading to the *R* absolute configuration with the angular methyl pointing downwards. Then, from this intermediate, the sulfinyl group can spontaneously be *beta*-*cis* eliminated producing sulfenic acid, which self-condensed to *p*-tosyl disulfide oxide and of course producing the desired product [23].

Because this methodology gives access to enantiopure sulfinylquinones, we applied it to benzoquinone **8** (Figure 6). First, compound **8** was engaged in a metalation and sulfinylation with the freshly prepared chiral (−)-*R*(*S*)-menthyl-*p*-toluenesulfinate **14** [28], leading to compound **15** with 47% yield. Then, direct oxidative demethylation using CAN afforded **16** in excellent yield. Finally, the Kochi–Anderson reaction [4] was performed using the 4-bromophenylacetic acid in a model reaction giving the sulfinylquinone **17** with 31% yield. We then tried to reproduce the Diels–Alder cycloaddition using the same diene and the reported condition [23]. However, our attempts were not successful using three different dienes: 1,3-butadiene, trimethylsilyl-1,3-butadiene and sulfolene. The reaction only led to degradation or there was no reaction at all (Appendix A). The non-reactivity of the sulfinylquinone **17** can be explained by the fact that the benzyl chain is positioned on the opposite side to that occupied by the *p*-tosyl arm of the sulfoxide, creating steric hindrance on both sides of the dienophile to prevent the approach of the diene (see TS, Figure 6).

### 2.3. Preliminary Investigations on the Synthesis of Ψ-Plasmodiones from Boronic Acid-Substituted 3-Benzyl-benzoquinone

We then decided to perform the Kochi–Anderson reaction [4] from compound **5** before the Diels–Alder reaction. However, the radical condition was not compatible with the boronic benzoquinone. In fact, Baran’s team [29] described a C-H functionalization on a quinone under the Kochi–Anderson condition, with arylboronic acid via a nucleophilic radical addition, which could potentially create in our case a fast polymerization chain reaction or the homocoupling of **5** happening before the initial Kochi–Anderson (Figure 7, Route A).

However, this limiting step opens possibilities to introduce the boronic acid after the benzylation of 3,6-dimethyl-2-benzoquinone **7** (Figure 7, Route B) and 3,5-dimethyl-2-bromobenzoquinone **18** (Figure 7, Route C) and potentially to obtain the desired Ψ-plasmodione regioisomers after a final Diels–Alder reaction. We thus synthesized the new precursors **21** and **22**, from compounds **19** and **20**, respectively, for the borylation, according to the reaction sequence shown in Figure 7 (similarly to Figure 4), with an overall yield of 42% and 24% from both commercial benzoquinones in three steps.

Oxidative demethylation of compound **23** with CAN at 0 °C gave a 2:1.5 mixture of boronic quinone **25** and the product of quinone dimerization **26**. These two products are easily separated by precipitation of the boronic quinone in CH_2_Cl_2_. Of note, Veguillas et al. also described the formation of a dimer (4,4′-dimethyl-1,1′-bicyclohexa-3,6-diene-2,2′,5,5′-tetraone) in an almost quantitative yield during CAN oxidation of 4-methyl-2,5-dimethoxyphenyl boronic acid [20]. This is made via an oxidative radical coupling, likely due to the initial deboronation of starting boronic acid, followed by coupling of the intermediate radicals and evolution to the final product by oxidative demethylation [30].

### 2.4. Synthesis of Ψ-Plasmodiones from Boronic Acid-Substituted 3-Benzyl-benzoquinones

Finally, we introduced the boronic group after the Kochi–Anderson reaction and performed the Diels–Alder cycloaddition at the end. Beginning with the 4a-Ψ-plasmodione regioisomer, compound **7** was engaged in a Kochi–Anderson reaction with 4-trifluoromethylpheneacetic acid producing compound **19** in good yield. Then, as for the synthesis of compound **5** (Figure 4), the bromo-benzoquinone was protected giving compound **21** and a subsequent metalation followed by a borylation was performed, which led to the formation of the boronic acid **23** after hydrolysis. The corresponding boronic acid was directly engaged in the oxidative demethylation with CAN producing the boronic-benzoquinone **25** with 40% yield. Finally, the desired 4a-Ψ-plasmodione **1** (10.5%, six steps) containing the angular methyl, was obtained through the Diels–Alder reaction (Figure 8). The same Diels–Alder cycloaddition condition used in Carreňo’s group [20,22] was used to generate the 4a-Ψ-plasmodione **1** in an excellent yield, producing the same regioisomer with the *endo* attack. By comparison with the sulfinyl methodology, we assumed that the Diels–Alder reaction was possible since the boronic acid is far less hindered. Therefore, the pendant benzyl of **25** did not block the attack of the diene or modified its approach. Furthermore, we applied the same methodology to generate the other regioisomer, the 8a-Ψ-plasmodione **2** (4.1%, 6 steps) from **20** (Figure 8).

In conclusion, we investigated different methodologies to produce non-quinonoid Ψ-plasmodione regioisomers substituted with an angular methyl. The main limitation comes from the absence of reactivity of sulfinylquinones under Diels–Alder reaction condition due to steric hindrance. However, with boronic acids as auxiliary to govern the regioselectivity of the cycloaddition and by choosing the right order of chemical steps, we were able to overcome these limitations and produce the desired regioisomer products (Figure 8). It should be noted that the presence of boron in boron-based prodrugs is known to significantly increase the therapeutic activity of final molecules [31], e.g., against parasitic protozoa, but this strategy has not yet been exploited in this work.

### 2.5. Antiparasitic Activities and Toxicity of Ψ-Plasmodione Regioisomers

The antimalarial and the antischistosomal activities of both synthesized Ψ-plasmodione regioisomers were measured against *P. falciparum* strain NF54 using the [^3^H]-hypoxanthine incorporation assay [32], against *S. mansoni* newly transformed schistosomula (NTS) [33] and against the ex vivo adult worms. Both quinone-free variants displayed low micromolar IC_50_ activity values around 2 µM against *P. falciparum*, values that are high in comparison with the potent antimalarial activity expressed by plasmodione (Table 1). The antischistosomal activities of both compounds was found relevant at 10 µM in the NTS, encouraging the determination of IC_50_ values both in the NTS and adult worm assays (Table 2). The antischistosomal potencies could suggest that a more effective prodrug effect took place in *S. mansoni*, both to release plasmodione and/or to express better properties by the non-aromatic Ψ-PD analogues, e.g., a better penetration of Ψ-plasmodiones across the tegument of the *S. mansoni* worms versus the membranes of the *Plasmodium falciparum*-parasitized red blood cells [34]. The in vivo activity of the most potent regioisomer 4a-Ψ-PD **1** has been tested in *S. mansoni*-infected mice (Table 3). However, 4a-Ψ-PD **1** showed a low activity with a worm burden reduction of 15.7%.

### 2.6. Metabolic Studies

We have previously shown that the *P. falciparum* glutathione reductase-catalyzed redox-cycling assay in the presence of PDO and methemoglobin generates a ferryl hemoglobin in a 3 h-long cascade of redox reactions [7]. More recently, we also demonstrated that the *P. falciparum* ferredoxin-NADP^+^ reductase (*Pf*FNR) efficiently catalyzes the benzylic oxidation of plasmodione in a 2 h-long lasting redox-cycling assay [6]. Here, we established a modified redox-cycling assay to study the oxidation of Ψ-plasmodiones by heme under (per)ferryl state, which had been generated by the NADPH-dependent *Pf*FNR catalyzed redox cycling of PDO (Figure 9). Of note, the reaction of heme(Fe^3+^) with moderated excess of hydrogen peroxide has been recently demonstrated to affect the oxidation state of heme’s iron with only a minimal degradation of the macrocyclic ring [35]. Changes in the oxidation state of the iron ion were suggested to be associated to the formation of ferryl heme species, i.e., the formation of Fe^IV^=O or Fe^IV^-OH ferryl species or a Fe^V^=O perferryl species [12,14] that represent a hyperoxidized state of the iron ion. Hence, the cascade of redox reactions induced by *Pf*FNR-catalyzed PDO reduction in the presence of hematin in open air might form (per)ferryl heme species continuously as soon as NADPH is added in the superoxide anion radical-generating assay.

During 3 h-long lasting *Pf*FNR-catalyzed PDO redox cycling subjected to repetitive addition of NADPH, hematin(Fe^3+^), NADPH and oxygen, we hypothesized that the angular methyl of the Ψ-plasmodiones could be oxidized and then release plasmodione after cleavage of the C-C bond and aromatization. To validate whether plasmodione is generated in the redox reaction mixture under several conditions, we analyzed the content by LC-ESI-MS. As shown in Figure 1, we could clearly detect the formation of plasmodione at 25 °C and 37 °C. In the MS spectra of the chromatography peaks at RT = 48.7 min and 46.3 min were observed at *m/z* 331.0941 and 345.07, assigned to the species PD ([M + H]^+^ *m/z* = 331.09) and PDO_ox_ ([M + H]^+^ *m/z* = 345.07). The exact mass of plasmodione was detected with an error of 0.1 ppm, attesting for the accurate assignment of the *m/z* peak to PD. In order to quantify the concentration of plasmodione generated in the 3 h course, we determined the area and intensity of *m/z* peaks after injection of plasmodione solutions at known concentrations: 1, 2, 5, 10 µM (Appendix A). With the calibration plots, we determined the concentration of plasmodione generated in the reaction mixtures. All reaction mixtures involving 4a-Ψ-plasmodione **1** or 8a-Ψ-plasmodione **2**, carried out at 25 °C or 37 °C, in the presence of hematin and the *Pf*FNR-catalyzed PDO redox-cycling system, exhibited the presence of plasmodione (RT = 48.7 min) at a concentration of 1.91 and 0.95 µM at 25 °C, respectively, or 4.04 and 2.65 µM at 37 °C, respectively (Appendix A).

## 3. Materials and Methods

### 3.1. Chemistry: General

All the reagents and solvents were purchased from commercial sources and used as received, unless otherwise stated. The ^1^H, ^19^F {^1^H} and, and ^13^C {^1^H} NMR spectra were obtained in CDCl_3_, Acetone-d_6_ as solvents using a 300 MHz, 400 MHz or 500 MHz spectrometer. Chemical shifts were reported in parts per million (δ). ^1^H NMR data were reported as follows: chemical shift (δ ppm) (multiplicity, coupling constant (Hz), and integration). Multiplicities are reported as follows: s = singlet, d = doublet, t = triplet, q = quartet, m = multiplet, or combinations thereof. High-resolution mass spectroscopy (HRMS) spectra were recorded using the electron spray ionization (ESI) technique. Reactants were purchased from commercial sources, such as Fluorochem (Hadfield, Derbyshire, UK), Sigma-Aldrich (Saint Quentin Fallavier, France), BLDpharm (Reinbek, Germany) and Alfa Aesar (Karlsruhe, Germany).

### 3.2. Synthesis of Precursors

*2-bromo-3,5-dimethylbenzene-1,4-diol* (**10**). 2,6-dimethylbenzoquinone (0.5 g, 3.7 mmol, 1 equiv.) was solubilized in Et_2_O (45.8 mL), then a solution of Na_2_S_2_O_4_ (4.8 g, 27.5 mmol, 7.5 equiv.) in water (45.8 mL) was added dropwise. The mixture was stirred 1 h at 25 °C. The crude product was extracted three times with CH_2_Cl_2_, washed with water, dried over magnesium sulphate, then solvent was removed under vacuum. The white solid was directly engaged in the next following step. The crude product was solubilized in acetonitrile (16.1 mL) and NBS (0.65 g, 3.62 mmol, 1.07 equiv.) was added. The mixture was stirred 16 h at 25 °C. The organic solvent was removed under reduced pressure and the crude product was purified by silica gel chromatography (CHX/EtOAc, 7/3, *v*/*v*, UV) to obtain 2-bromo-3,5-dimethylbenzene-1,4-diol (290.9 mg, 36.5%) as a white solid. ^1^H NMR (400 MHz, CDCl_3_) *δ* 6.72 (s, 1H), 5.14 (s, 1H), 4.30 (s, 1H), 2.34 (s, 3H), 2.20 (s, 3H). In accordance with a previously published method [22].

*2-bromo-1,4-dimethoxy-3,5-dimethylbenzene* (**11**). Compound **10** (290 mg, 1.34 mmol, 1 equiv.) was solubilized in acetone (13.4 mL) then K_2_CO_3_ (1.66 g, 12 mmol, 9 equiv.) and Me_2_SO_4_ (0.76 mL, 8 mmol, 6 equiv.) were added. The mixture was stirred under reflux 60 °C for 4 h. A 1M NaOH aqueous solution was added, then the organic solvent was removed under reduced pressure, extracted three times with CH_2_Cl_2_, washed with water, dried over magnesium sulphate and the solvent was removed under vacuum. The yellow oil was directly engaged in the next reaction. ^1^H NMR (400 MHz, CDCl_3_) *δ* 6.60 (s, 1H), 3.85 (s, 3H), 3.66 (s, 3H), 2.37 (s, 3H), 2.27 (s, 3H). In accordance with a previously published method [22].

*(−)-R(S)-menthyl-p-toluenesulfinate* (**14**). Hydrated sodium *p*-tolylsulfinate (200 g) salt was previously dried by azeotropic distillation with toluene (500 mL) during 24 h. Dried sodium *p*-toluenesulfinate (50 g, 0.28 mmol, 1 equiv.) was added portionwise to a solution of thionyl chloride (50 mL, 0.69 mmol, 2.5 equiv.) in toluene (100 mL) at 0 °C. The mixture was stirred at room temperature for 1 h after the end of the addition. The reaction mixture was concentrated by distillation of the azeotrope SOCl_2_/toluene under reduced pressure. The resulting oil was dissolved in anhydrous diethyl ether (130 mL) and the white slurry was cooled to 0 °C. At this temperature, a solution of (−)-menthol (43.8 g, 0.28 mmol, 1 equiv.) in pyridine (50 mL) was added dropwise. After the end of the addition, the mixture was stirred 2 h at 25 °C. The reaction was slowly quenched at 0 °C by distilled water (100 mL). The two phases were separated and the organic phase was washed with a 1 M HCl solution, washed with saturated NaCl solution (OR) brine, dried over magnesium sulphate and the solvent was removed under vacuum. The crude product was dissolved in acetone (100 mL) and a few drops of concentrated HCl solution were added. The resulting mixture was allowed to crystallize in the freezer. The resulting crystals were filtered and washed with cold hexane. The mother solution was concentrated and the same operation was repeated several times giving the desired product (75.2 g, 90%), as white crystal. M.p. = 110 °C. [α${]}_{25}^{D}$ = −202. ^1^H NMR (300 MHz, CDCl_3_) *δ* 7.46 (m, *J* = 8.2 Hz, 4H), 4.12 (td, *J* = 11.0 Hz, *J* = 4.5 Hz, 1H), 2.40 (s, 3H), 2.28 (dtd, *J* = 12.1 Hz, *J* = 4.5 Hz, *J* = 1.9 Hz, 1H), 2.13 (sept d, *J* = 6.9 Hz, *J* = 2.6 Hz, 1H), 1.74–1.64 (m, 2H), 1.48 (m, 1H), 1.35 (ddd, *J* = 11.0, *J* = 3.2 Hz, *J* = 2.6 Hz, 1H), 1.22 (td, *J* = 12.1 Hz, *J* = 11.0 Hz, 1H), 1.04 (dddd, *J* = 15.5 Hz, *J* = 11.0 Hz, *J* = 2.6 Hz, *J* = 1.1 Hz, 1H), 0.96 (d, *J* = 6.5 Hz, 3H), 0.92–0.82 (m, 1H), 0.86 (d, *J* = 6.9 Hz, 3H), 0.72 (d, *J* = 6.9 Hz, 3H). ^13^C {^1^H} NMR (75 MHz, CDCl_3_) *δ* 143.2, 142.4, 129.6, 125.0, 80.1, 47.9, 42.9, 31.7, 25.2, 23.2, 22.0, 21.5, 20.8, 15.5. In accordance with a previously published method [28].

*(S)-1,4-dimethoxy-2,5-dimethyl-3-(p-tolylsulfinyl)benzene* (**15**). 2-Bromo-1,4-dimethoxy-3,5-dimethylbenzene **8** (2.5 g, 10.2 mmol, 1 equiv.) in 5 mL THF was added dropwise to solid magnesium (261.3 mg, 10.7 mmol, 1.05 equiv.) in 10 mL THF under argon. Dibromoethane (0.1 mL) was then added to initiate the reaction and the mixture was stirred for 2 h at 25 °C. It was then cooled at 0 °C and added dropwise to a solution of (−)-*R*(*S*)-menthyl-*p*-toluenesulfinate **14** (3.9 g, 13.3 mmol, 1.3 equiv.). The mixture was allowed to warm to 25 °C and stirred overnight. It was then cooled to 0 °C and saturated NH_4_Cl was added. The crude product was extracted with diethyl ether, washed with brine and dried over magnesium sulphate, then the solvent was removed under vacuum. The crude product was finally purified by silica gel chromatography (CHX/EtOAc, 8/2, *v*/*v*, UV) to obtain the desired product (1.47 g, 47%) as a white solid. ^1^H NMR (300 MHz, CDCl_3_) *δ* 7.45–7.47 (d, *J* = 8.23 Hz, 2H), 7.23–7.25 (d, *J* = 8.06 Hz, 2H), 6.78 (s, 1H), 3.83 (s, 3H), 3.78 (s, 3H), 2.37 (s, 3H), 2.31 (s, 3H), 2.29 (s, 3H). [α${]}_{20}^{D}$ = −180.46. In accordance with a previously published method [36].

### 3.3. General Procedure of the Bromination

To a solution of the corresponding substrate (1 equiv.) in CH_2_Cl_2_ (0.36 M), a solution of bromine (1.05 equiv.) in CH_2_Cl_2_ (1.26 M) was added dropwise at 0 °C. The mixture was stirred for 2h at 25 °C, when DIPEA (1.0 equiv.) was added. The mixture was stirred for 3 h at 25 °C and poured into water. Water was added and the crude product extracted three times with CH_2_Cl_2_, washed with water and dried over magnesium sulphate, then the solvent was removed under vacuum. The crude product was finally purified by silica gel chromatography with the appropriate solvent.

*3-bromo-2,5-dimethylcyclohexa-2,5-diene-1,4-dione* (**7**). With 2,5-dimethylbenzoquinone, eluent (CHX/EtOAC, 9/1, *v*/*v*, UV), yellow solid, 77% yield. ^1^H NMR (400 MHz, CDCl_3_) *δ* 6.63 (q, *J* = 1.6 Hz, 1H), 2.19 (s, 3H), 2.10 (d, *J* = 1.6 Hz, 3H). ^13^C {^1^H} NMR (101 MHz, CDCl_3_) *δ* 184.3, 179.9, 146.1, 145.7, 136.0, 133.2, 16.9, 16.6. In accordance with a previously published method [37].

*2-bromo-3,5-dimethylcyclohexa-2,5-diene-1,4-dione* (**18**). With 2,6-dimethylbenzoquinone, eluent (T, UV), orange solid, 47.5% yield. ^1^H NMR (400 MHz, CDCl_3_) *δ* 6.76 (q, *J* = 1.6 Hz, 1H), 2.24 (s, 3H), 2.08 (d, *J* = 1.6 Hz, 3H). ^13^C {^1^H} NMR (101 MHz, CDCl_3_) *δ* 184.3, 179.9, 146.1, 145.7, 136.0, 133.2, 16.9, 16.6. HRMS (ESI) calcd. for C_8_H_8_BrO_2_: 214.9702. Found: 214.9709 ([M + H]^+^). In accordance with a previously published method [37].

### 3.4. General Procedure of the Reduction and Protection of Benzoquinone

The corresponding bromo-benzoquinone (1 equiv.) was solubilized in MeOH (0.26 M), then a solution of SnCl_2_ (2.5 equiv.) in 37% HCl (4.12 equiv.) was added dropwise and the mixture was stirred 30 min at 25 °C until the solution came back yellowish. Most of the solvent was evaporated under vacuum and the white precipitate was rinsed with water. The powder was dissolved in acetone (0.26 M) and dry over magnesium sulfate. Argon dimethyl sulfate (5 equiv.) was added to the previous mixture, then a solution of KOH (5 equiv.) in MeOH (1 M) was added dropwise. When the addition was completed, the mixture was stirred under reflux at 60 °C during 3 h. A 20% KOH aqueous solution (10 mL) was added to the mixture and organic solvent was removed under reduced pressure. The crude product was extracted three times with CH_2_Cl_2_, washed with water and dried over magnesium sulphate, then the solvent was removed under vacuum. The crude product was finally purified by silica gel chromatography with the appropriate solvent.

*2-bromo-1,4-dimethoxy-3,5-dimethylbenzene* (**8**). With compound **7**, eluent (CHX/EtOAc, 8/2, *v*/*v*, UV), white solid, 80% yield. M.p. = 56–58 °C. ^1^H NMR (300 MHz, CDCl_3_) *δ* 6.60 (s, 1H), 3.78 (s, 3H), 3.74 (s, 3H), 2.31 (s, 3H), 2.28 (s, 3H). ^13^C {^1^H} NMR (75 MHz, CDCl_3_) *δ* 152.2, 151.1, 132.6, 130.1, 111.8, 111.4, 60.3, 56.5, 16.6, 16.4. HRMS (ESI) calcd. for C_10_H_13_BrO_2_: 244.0099. Found: 244.0108 ([M]^+^). In accordance with a previously published method [22].

*1-bromo-2,5-dimethoxy-4,6-dimethyl-3-(4-(trifluoromethyl)benzyl)benzene* (**22**). With compound **20**, eluent (T/CHX, 5/5, *v*/*v*, UV), colorless oil, 73.6% yield. ^1^H NMR (400 MHz, CDCl_3_) *δ* 7.49 (d, *J* = 8.1 Hz, 2H), 7.20 (d, *J* = 8.2 Hz, 2H), 4.15 (s, 2H), 3.69 (s, 3H), 3.66 (s, 3H), 2.40 (s, 3H), 2.08 (s, 3H). ^19^F {^1^H} NMR (377 MHz, CDCl_3_) *δ* −62.34. ^13^C {^1^H} NMR (101 MHz, CDCl_3_) *δ* 154.0, 152.2, 144.5 (q, *J* = 1.4 Hz), 131.6, 130.9, 130.5, 129.2, 128.45, 128.44 (q, *J* = 24.5 Hz), 128.3, 125.5 (q, *J* = 3.7 Hz), 124.4 (q, *J* = 276 Hz), 118.3, 61.2, 60.5, 33.3, 16.9, 12.9. HRMS (ESI) calcd. for C_18_H_18_BrO_2_F_3_: 402.0437. Found: 402.0418 ([M + H]^+^).

*1-bromo-2,5-dimethoxy-3,6-dimethyl-4-(4-(trifluoromethyl)benzyl)benzene* (**21**). With compound **19**, eluent (T/CHX, 5/5, *v*/*v*, UV), colorless oil, 75% yield. ^1^H NMR (400 MHz, CDCl_3_) *δ* 7.49 (d, *J* = 8.1 Hz, 2H), 7.19 (d, *J* = 7.9 Hz, 2H), 4.10 (s, 2H), 3.75 (s, 3H), 3.57 (s, 3H), 2.39 (s, 3H), 2.14 (s, 3H). ^19^F {^1^H} NMR (377 MHz, CDCl_3_) *δ* −62.34. ^13^C {^1^H} NMR (101 MHz, CDCl_3_) *δ* 153.9, 152.3, 144.4 (q, *J* = 1.4 Hz), 131.3, 130.3, 130.0, 129.2, 128.4, 128.5 (q, *J* = 24.5 Hz),128.4, 125.5 (q, *J* = 3.1 Hz), 124.4 (q, *J* = 276 Hz), 119.7, 61.3, 60.4, 32.8, 17.1, 13.3. HRMS (ESI) calcd. for C_18_H_19_BrO_2_F_3_: 403.0515. Found: 403.0537 ([M + H]^+^).

### 3.5. General Procedure of the Metalation and Borylation of the Protected Bromo-Benzoquinone

A solution of *n*-BuLi (1.2 equiv.) (1.6 M in hexane) was added dropwise to a stirred solution of the corresponding protected bromo-benzoquinone (1 equiv.) in anhydrous THF (0.17 M) under argon at −78 °C. The mixture was stirred at −80 °C for 5 min and then triisopropyl borate (2.5 equiv.) was added dropwise at −78 °C. The mixture was stirred back to 25 °C for 16 h and quenched with HCl 1 M, the crude product was extracted three times with EtOAc, washed with water and dried over magnesium sulphate, then the solvent was removed under vacuum. The crude product was finally purified by silica gel chromatography with the appropriate solvent.

*(2,5-dimethoxy-3,6-dimethylphenyl)boronic acid* (**9**). With compound **8**, no purification, white solid, 55% NMR yield. ^1^H NMR (300 MHz, Acetone-*d*_6_) *δ* 6.68 (s, 1H), 3.76 (s, 3H), 3.68 (s, 3H), 2.20 (s, 3H), 2.13 (s, 3H). In accordance with a previously published method [26].

*(3,6-dimethoxy-2,4-dimethylphenyl)boronic acid* (**12**). With compound **11**, eluent (CHX/EtOAc, 6/4, *v*/*v*, UV), white solid, 54% yield over two steps. ^1^H NMR (400 MHz, CDCl_3_) *δ* 6.58 (s, 1H), 5.83 (s, 2H), 3.83 (s, 3H), 3.65 (s, 3H), 2.48 (s, 3H), 2.31 (s, 3H). In accordance with a previously published method [22].

*(2,5-dimethoxy-4,6-dimethyl-3-(4-(trifluoromethyl)benzyl)phenyl)boronic acid* (**24**). With compound **22**, eluent (CHX/EtOAc, 5/5, *v*/*v*, UV), white solid, 45.5% yield, contains a small impurity. M.p. = 115–117 °C. ^1^H NMR (400 MHz, CDCl_3_) *δ* 7.49 (d, *J* = 8.1 Hz, 2H), 7.18 (d, *J* = 8.0 Hz, 2H), 5.93 (s, 2H), 4.08 (s, 2H), 3.65 (s, 3H), 3.62 (s, 3H), 2.44 (s, 3H), 2.11 (s, 3H). ^19^F {^1^H} NMR (377 MHz, CDCl_3_) *δ* −62.32. ^13^C {^1^H} NMR (101 MHz, CDCl_3_) *δ* 159.3, 154.2, 144.8 (q, *J* = 1.5 Hz), 135.9, 134.1, 129.0, 128.5 (q, *J* = 24.5 Hz), 128.4, 128.4, 125.4 (q, *J* = 3.8 Hz), 124.4 (q, *J* = 276 Hz), 62.7, 60.1, 32.4, 15.6, 13.1. HRMS (ESI) calcd. for C_18_H_19_BO_4_F_3_: 367.1334. Found: 367.1331 ([M + H]^+^).

*(2,5-dimethoxy-3,6-dimethyl-4-(4-(trifluoromethyl)benzyl)phenyl)boronic acid* (**23**). With compound 21, yellow oil, 63% NMR yield, not purified; engaged in the next step.

### 3.6. General Procedure of the Oxidative Demethylation of the Protected Benzoquinone

The corresponding protected benzoquinone (1 equiv.) was solubilized in acetonitrile (0.08 M), then a solution of CAN (2.2 equiv.) in water (0.24 M) was added. The mixture was stirred 1 h at 25 °C. The organic solvent was removed under reduced pressure and the crude product extracted three times with CH_2_Cl_2_, washed with water and dried over magnesium sulphate, then the solvent was removed under vacuum. The crude product was finally purified by silica gel chromatography with the appropriate solvent.

*(2,5-dimethyl-3,6-dioxocyclohexa-1,4-dien-1-yl)boronic acid* (**5**). With compound **9**, no purification, yellow solid, 99% yield. ^1^H NMR (300 MHz, CDCl_3_) *δ* 6.64 (s, 2H), 6.58–6.59 (q, *J* = 1.52 Hz, 1H), 2.30 (s, 3H), 1.98 (d, *J* = 1.53 Hz, 3H). In accordance with a previously published method [22].

*(2,4-dimethyl-3,6-dioxocyclohexa-1,4-dien-1-yl)boronic acid* (**6**). With compound **12**, no purification, yellow solid, 91.5% yield. ^1^H NMR (400 MHz, CDCl_3_) *δ* 7.00 (s, 2H), 6.60 (q, *J* = 1.6 Hz, 1H), 2.41 (s, 3H), 2.07 (d, *J* = 1.6 Hz, 3H). In accordance with a previously published method [22].

*(S)-2,5-dimethyl-3-(p-tolylsulfinyl)cyclohexa-2,5-diene-1,4-dione* (**16**). With compound **15**, no purification, red solid, 95% yield. ^1^H NMR (300 MHz, CDCl_3_) *δ* 7.64–7.67 (d, *J* = 8.24 Hz, 2H), 7.31–7.34 (d, *J* = 8.00 Hz, 2H), 6.65 (d, *J* = 1.54 Hz, 1H), 2.50 (s, 3H), 2.42 (s, 3H), 2.03 (d, *J* = 1.54 Hz, 3H). [α${]}_{20}^{D}$ = +602.5. In accordance with a previously published method [36].

*(2,4-dimethyl-3,6-dioxo-5-(4-(trifluoromethyl)benzyl)cyclohexa-1,4-dien-1-yl)boronic acid* (**27**). With compound **24**, eluent (T/CHX, 7/3, *v*/*v*, UV), yellow solid, 63.8% yield, with the same small impurity as observed for **24**. M.p. = 89–91 °C. ^1^H NMR (400 MHz, CDCl_3_) *δ* 7.53 (d, *J* = 8.2 Hz, 2H), 7.28 (d, *J* = 7.8 Hz, 2H), 6.77 (s, 2H), 3.91 (s, 2H), 2.40 (s, 3H), 2.13 (s, 3H). ^19^F {^1^H} NMR (377 MHz, CDCl_3_) *δ* −62.51. ^13^C {^1^H} NMR (101 MHz, CDCl_3_) *δ* 193.8, 188.0, 157.5 (q, *J* = 1.5 Hz), 143.2, 142.9, 128.94, 128.91, 128.8 (q, *J* = 24.5 Hz), 125.8 (q, *J* = 4.0 Hz), 32.1, 15.7, 13.1. HRMS (ESI) calcd. for C_16_H_13_BO_4_F_3_: 337.0864. Found: 337.0866 ([M + H]^+^).

*(2,5-dimethyl-3,6-dioxo-4-(4-(trifluoromethyl)benzyl)cyclohexa-1,4-dien-1-yl)boronic acid* (**25**). With compound **23**, no purification, yellow solid, 40% yield. ^1^H NMR (300 MHz, CDCl_3_) *δ* 7.41 (m, *J* = 8.04, 4H), 6.76 (s, 2H), 3.94 (s, 2H), 2.40 (s, 3H), 2.11 (s, 3H). ^19^F {^1^H} NMR (376 MHz, CDCl_3_) *δ* −62.51. ^13^C {^1^H} NMR (101 MHz, CDCl_3_) *δ* 194.5, 187.5, 157.0, 155.3, 143.2, 143.1, 142.9, 142.6, 142.2, 128.9, 125.7, 125.6, 122.9, 32.2, 17.2, 15.5.

### 3.7. General Procedure of the Kochi–Anderson Reaction

The corresponding benzoquinone (1 equiv.) and the corresponding phenyl acetic acid (2 equiv.) were dissolved in a mixture of acetonitrile (0.06 M) and water (0.2 M). Then, AgNO_3_ (0.35 equiv.) and ammonium persulfate (1.3 equiv.) were added in the reaction mixture. The yellow mixture was protected from light and stirred under reflux for 3 h. The organic solvent was removed under reduced pressure and the crude product extracted three times with CH_2_Cl_2_, washed with water and dried over magnesium sulphate, then the solvent was removed under vacuum. The crude product was finally purified by silica gel by silica gel chromatography with the appropriate solvent.

*2-bromo-3,6-dimethyl-5-{[4-(trifluoromethyl)phenyl]methyl}cyclohexa-2,5-diene-1,4-dione* (**19**). With compound **7** and 4-(trifluomethylphenyl)acetic acid, eluent (T/CHX, 7/3, *v*/*v*, UV), yellow oil, 77% yield. ^1^H NMR (400 MHz, CDCl_3_) *δ* 7.53 (d, *J* = 8.1 Hz, 2H), 7.29 (d, *J* = 8.3 Hz, 2H), 3.94 (s, 2H), 2.23 (s, 3H), 2.18 (s, 3H). ^19^F {^1^H} NMR (377 MHz, CDCl_3_) *δ* −62.49. ^13^C {^1^H} NMR (101 MHz, CDCl_3_) *δ* 183.9, 179.7, 145.9, 142.2, 142.1, 141.8 (q, *J* = 1.5 Hz), 135.9, 128.97 (q, *J* = 24.2 Hz), 128.94, 125.7 (q, *J* = 3.8 Hz), 124.2 (q, *J* = 279.6 Hz), 32.3, 17.3, 13.8. HRMS (ESI) calcd. for C_16_H_13_BrO_2_F_3_: 373.0046. Found: 373.0040 ([M + H]^+^).

*2-bromo-3,5-dimethyl-6-(4-(trifluoromethyl)benzyl)cyclohexa-2,5-diene-1,4-dione* (**20**). With compound **18** and 4-(trifluomethylphenyl)acetic acid, eluent (T/CHX, 7/3, *v*/*v*, UV), yellow solid, 65.7% yield. M.p. = 67–69 °C. ^1^H NMR (400 MHz, CDCl_3_) *δ* 7.52 (d, *J* = 8.1 Hz, 2H), 7.30 (d, *J* = 7.9 Hz, 2H), 3.97 (s, 2H), 2.24 (s, 3H), 2.15 (s, 3H). ^19^F {^1^H} NMR (377 MHz, CDCl_3_) *δ* −62.53. ^13^C {^1^H} NMR (101 MHz, CDCl_3_) *δ* 184.6, 179.3, 146.3, 142.5, 141.9, 141.8 (q, *J* = 1.6 Hz), 135.7, 129.14 (q, *J* = 24.2 Hz), 129.06, 125.8 (q, *J* = 3.8 Hz), 124.3 (q, *J* = 279.6 Hz), 32.9, 17.3, 13.3. HRMS (ESI) calcd. for C_16_H_12_O_2_^79^BrF_3_^23^Na: 394.9865. Found: 394.9860 ([M + Na]^+^).

*(S)-2-(4-bromobenzyl)-3,6-dimethyl-5-(p-tolylsulfinyl)cyclohexa-2,5-diene-1,4-dione* (**17**). With compound **16** (S)-2,5-dimethyl-3-(*p*-tolylsulfinyl)cyclohexa-2,5-diene-1,4-dione (400 mg, 1.46 mmol, 1 equiv.) and 2-(4-bromophenyl)acetic acid (627.9 mg, 2.92 mmol, 2 equiv.), eluent (T/Et_2_O, 95/5, *v*/*v*, UV), red solid, 31% yield. M.p. = 111–113 °C. ^1^H NMR (300 MHz, CDCl_3_) *δ* 7.66 (d, *J* = 8.22 Hz, 2H), 7.40 (d, *J* = 8.35 Hz, 2H), 7.32 (d, *J* = 8.19 Hz, 2H), 7.02 (d, *J* = 8.31 Hz, 2H), 3.79 (s, 2H), 2.50 (s, 3H), 2.41 (s, 3H), 2.08 (s, 3H). ^13^C {^1^H} NMR (75 MHz, CDCl_3_) *δ* 185.7, 184.5, 146.7, 145.9, 143.2, 141.8, 141.6, 139.6, 136.3, 131.8, 130.3, 130.1, 124.9, 120.6, 31.9, 21.4, 12.5, 9.4. [α${]}_{20}^{D}$ = 337.5.

### 3.8. General Procedure of the Diels–Alder Cycloaddition Reaction

The corresponding boronic acid (1 equiv.) was solubilized in CH_2_Cl_2_ (0.07 M), then (3E/Z)-4-methoxybuta-1,3-diene (3 equiv.) was added at −20 °C and the mixture was stirred for 1 h at −20 °C. Water was added and the crude product was extracted three times with CH_2_Cl_2_, washed with water and dried over magnesium sulphate, then the solvent was removed under vacuum. The crude product was finally purified by silica gel chromatography with the appropriate solvent.

*(±)-2,4a-dimethyl-4a,5-dihydronaphthalene-1,4-dione* (**3**). With compound **5**, no purification, yellow oil, 99% yield. ^1^H NMR (300 MHz, CDCl_3_) *δ* 7.05 (m, 1H), 6.56 (m, 1H), 6.20 (m, 2H), 2.57 (ddd, *J* = 19.1, 5.2, 1.4 Hz, 1H), 2.46 (m, 1H), 1.77 (t, *J* = 1.5 Hz, 3H), 1.17 (d, *J* = 1.4 Hz, 3H). In accordance with a previously published method [22].

*(±)-3,4a-dimethyl-4a,5-dihydronaphthalene-1,4-dione* (**13**). With compound **12**, eluent (CHX/EtOAc, 8/2, *v*/*v*, UV), yellow oil, 52.6% yield. ^1^H NMR (400 MHz, CDCl_3_) *δ* 7.06 (dt, *J* = 5.3, 1.1 Hz, 1H), 6.78 (q, *J* = 1.5 Hz, 1H), 6.33–6.18 (m, 2H), 2.64–2.56 (m, 2H), 2.06 (d, *J* = 1.5 Hz, 3H), 1.23 (s, 3H). In accordance with a previously published method [22].

*(±)-2,4a-dimethyl-3-(4-(trifluoromethyl)benzyl)-4a,5-dihydronaphthalene-1,4-dione* (**1**). With compound **25**, 2 fold-crystallized, yellow solid, 99% yield. M.p. = 59–61 °C (hexane). ^1^H NMR (500 MHz, CDCl_3_) *δ* 7.52 (d, *J* = 8.0 Hz, 2H), 7.28 (d, *J* = 8.0 Hz, 2H), 7.15 (dd, *J* = 5.4, 1.1 Hz, 1H), 6.32–6.22 (m, 2H), 4.08 (d, *J* = 14.2 Hz, 1H), 3.84 (d, *J* = 14.3 Hz, 1H), 2.60 (dd, *J* = 4.3, 1.7 Hz, 2H), 2.18 (s, 3H), 1.12 (s, 3H). ^19^F {^1^H} NMR (377 MHz, CDCl_3_) *δ* −62.49. ^13^C {^1^H} NMR (126 MHz, CDCl_3_) *δ* 200.3, 185.1, 148.1, 144.7, 142.1 (q, *J* = 1.6 Hz), 134.3, 133.5, 130.9, 128.9, 128.8 (q, *J* = 24 Hz), 125.8 (q, *J* = 3.7 Hz), 124.3 (q, *J* = 279.6 Hz), 123.6, 44.1, 33.1, 32.1, 24.9, 13.7. Elemental analysis calcd. for C_20_H_17_F_3_O_2_: C, 69.36; H, 4.95; found: C, 69.62; H, 5.08. HRMS (ESI) calcd. for C_20_H_18_O_2_F_3_: 347.1253. Found: 347.1244 ([M + H]^+^).

*(±)-3,4a-dimethyl-2-(4-(trifluoromethyl)benzyl)-4a,5-dihydronaphthalene-1,4-dione* (**2**). With compound **27**, eluent (CHX/EtOAc, 9/1, *v*/*v*, UV), yellow oil, 58.7% yield. ^1^H NMR (400 MHz, CDCl_3_) *δ* 7.52 (d, *J* = 8.1 Hz, 2H), 7.30 (d, *J* = 8.0 Hz, 2H), 7.15 (dd, *J* = 5.3, 1.2 Hz, 1H), 6.35–6.19 (m, 2H), 4.10–3.91 (m, 2H), 2.63 (dd, *J* = 4.6, 1.6 Hz, 2H), 2.12 (s, 3H), 1.21 (s, 3H). ^19^F {^1^H} NMR (377 MHz, CDCl_3_) *δ* −62.48. ^13^C {^1^H} NMR (101 MHz, CDCl_3_) *δ* 201.0, 184.4, 147.9, 144.9, 142.4 (q, *J* = 1.6 Hz), 134.3, 133.6, 131.0, 128.9, 128.8 (q, *J* = 24 Hz), 125.7 (q, *J* = 3.7 Hz), 124.3 (q, *J* = 279.6 Hz), 123.6, 44.2, 32.6, 32.2, 25.2, 14.3. HRMS (ESI) calcd. for C_20_H_18_O_2_F_3_: 347.1253. Found: 347.1255 ([M + H]^+^).

### 3.9. Parasite Culture and Antiplasmodial Drug Assays

*P. falciparum* NF54 wild type parasites cultured in medium containing 0.5% Albumax II were used to test for compound activity on parasite multiplication using a [^3^H]-hypoxanthine incorporation assay [32]. Compounds were dissolved in DMSO at 10 mM, serial dilutions prepared in hypoxanthine-free culture medium (7-step dilution series; 2-fold serial dilutions) and 100 µL aliquots were dispensed in duplicates into 96-well cell culture plates. The 100 µL asexual parasite culture suspensions (prepared in hypoxanthine-free medium) were added to each well and mixed with the preloaded compounds to obtain a final hematocrit of 1.25% and a parasitemia of ~0.3%. Each plate included eight wells containing the DMSO vehicle alone (0.1% final concentration) and four wells containing uninfected RBCs (uRBC). After incubation for 48 h, 0.25 μCi of [^3^H]-hypoxanthine was added per well and plates were incubated for an additional 24 h. Parasites were harvested onto glass-fiber filters using a Microbeta FilterMate cell harvester (Perkin Elmer, Waltham, MA, USA) and radioactivity was counted using a MicroBeta2 liquid scintillation counter (Perkin Elmer, Waltham, MA, USA). The results were recorded, processed by subtraction of the mean background signal obtained from the uRBC controls and expressed as a percentage of the mean signal obtained from the untreated controls [32,38]. Chloroquine diphosphate (Sigma C6628, Merck KGaA, Darmstadt, Germany) and artesunate (Mepha, Esch, Switzerland) were included as reference compounds in every experiment.

### 3.10. Drug Assay Against S. mansoni NTS

Harvested *S. mansoni* cercariae (Liberian strain) obtained from infected *Biomphalaria glabrata* snails were mechanically transformed into newly transformed schistosomula (NTS) following standard procedures as described [33]. Artesunate and praziquantel were previously tested in the drug assay against *S. mansoni* NTS [39]. About 30–40 NTS were placed in each well of a 96-well plate with culture medium and the test compound (25 μM, 10 μM, 1 μM) for a final well volume of 250 µL. Culture medium was composed of M199 medium (Gibco, Waltham MA, USA) supplemented with 5% Horse Serum (Gibco, Waltham MA, USA), 1% penicillin/streptomycin mixture (Invitrogen, Carlsbad, CA, USA, 100 U/mL) and 1% Mäser Mix [40]. Each compound was tested in triplicate. NTS incubated with no more than 1% DMSO served as control. NTS were kept in the incubator at 37 °C and 5% CO_2_ for up to 72 h. After 24, 48 and 72 h, the condition of the NTS was microscopically evaluated. Worms were scored as 0 = dead; 0.25–1 = reduced motility and significant tegument damage; 1.25–2 = reduced motility or marked tegument damages; 2.25–3 = viable, nice tegument, good motility.

### 3.11. Drug Assay Against Ex Vivo S. mansoni Adult Worms

Female NMRI mice (age 3 weeks, weight ca. 14–20 g) were purchased from Charles River (Sulzfeld, Germany). The animals were allowed to adapt for 1 week under controlled conditions (21–24 °C, 45–65% humidity, 12 h light, and free access to water and rodent diet) before experimental handling. To obtain adult schistosomes, NMRI mice were infected subcutaneously with 80 to 100 cercariae. After 49 days, the mice were euthanized with CO_2_ and the worms collected from the hepatic portal and mesenteric veins. Three pairs of adult worms were placed in each well of a 24-well plate with 2 mL culture medium and the test compound (25 μM, 10 μM). Culture medium was composed of RPMI 1640 (Invitrogen) supplemented with 5% Horse Serum (Gibco, Waltham, MA, USA) and 1% penicillin/streptomycin mixture (Invitrogen, 100 U/mL). Each compound was tested in duplicate. Artesunate and praziquantel were previously tested in the drug assay against *S. mansoni* adult worms [39]. Schistosomes incubation with no more than 1% DMSO served as control. Worms were kept in an incubator at 37 °C and 5% CO_2_ for up to 72 h. After 24, 48 and 72 h, the condition of the worms was microscopically evaluated and scored as described above [33].

### 3.12. Drug Assay In Vivo in S. mansoni-Infected Mice

The in vivo study was carried out in accordance with Swiss national and cantonal regulations on animal welfare under the permission number 545. Female mice (NMRI strain; 3-week-old; weight ca. 20–22 g) were purchased from Charles River, Germany. Rodents were kept under environmentally controlled conditions (temperature ~25 °C; humidity ~70%; 12 h light, 12 h dark cycle) with free access to water and rodent diet, and acclimatized for one week before infection. After 49 days post- infection, four mice were treated with a single oral dose of 4a-Ψ-PD **1** dissolved in a 7% (*v*/*v*) Tween 80 and 3% (*v*/*v*) ethanol aqueous solvent vehicle (10 mL/kg). Untreated mice (*n* = 4) served as controls in all experiments. Two weeks post-treatment mice were dissected and all worms counted. Worm burden reductions were calculated.

### 3.13. Cytotoxicity Assays with the Rat L6 Cell Line

Cell proliferation was assessed with resazurin, and the generally cytotoxic agent podophyllotoxin served as the positive control. L-6 cells, a primary cell line derived from rat skeletal myoblasts, were cultivated in RPMI 1640 medium supplemented with 1% L-glutamine (200 mM) and 10% fetal bovine serum. Cultures were maintained at 37 °C in an atmosphere of 5% CO_2_. Assays were performed in 96-well plates, each well containing RPMI 1640 medium supplemented with 1% L-glutamine (200 mM), 10% fetal bovine serum, and 2000 L-6 cells. Plates were incubated at 37 °C under a 5% CO_2_ atmosphere for 24 h. After that, compounds were dissolved in DMSO (10 mM or 10 mg/mL), serial drug dilutions of eleven 3-fold dilution steps covering a range from 100 to 0.002 μg/mL or 100 to 0.002 µM were prepared, and the plates incubated at 37 °C under 5% CO_2_ for 70 h. After 70 h of incubation, 10 µL of resazurin solution (12.5 mg resazurin in 100 mL 1xPBS) was added per well and plates were incubated for an additional 2 h. After that, the plates were read with a Spectramax Gemini EM microplate fluorometer (Molecular Devices, San Jose, CA, USA) using an excitation wave length of 536 nm and an emission wave length of 588 nm. The results were expressed as percentage of the untreated controls [41,42]. Fifty percent inhibitory concentrations (IC_50_) were estimated by linear interpolation [43].

### 3.14. PfFNR-Catalyzed PDO Redox-Cycling Assay with Ψ-Plasmodiones

The reaction mixtures in a total volume of 200 µL and final DMSO concentration of 10% in 50 mM PBS buffer pH 7.0 contained 50 µM of Ψ-PD, 13 µM hematin, and the *Pf*FNR-catalyzed PDO redox-cycling system consisting in 40 µM PDO, 100 µM NADPH and 0.5 µM *Pf*FNR. The reaction was started by addition of *Pf*FNR (4 µL) and sustained by subsequent addition of 100 µM of NADPH (in 2 µL) every 15 min for 3 h. Reactions were incubated at 25 °C or 37 °C for the time of redox cycling. For the controls, 80 µM H_2_O_2_ was added instead of the *Pf*FNR-catalyzed PDO redox-cycling system. The reaction was started by addition of *Pf*FNR (0.5 µM) and collected after 3 h incubation at 25 °C or 37 °C. Collected samples were centrifuged and diluted with 30 µL DMSO before LC-MS analyses. LC-MS analyses of the reaction mixtures were performed using an Agilent 1100 series LC coupled to a maXis II Q-TOF mass spectrometer (Bruker, Karlsruhe, Germany). *Pf*FNR-catalyzed PDO redox-cycling reaction mixtures containing hematin and each Ψ-plasmodione regioisomer was analyzed with the LC-MS system. The mass spectrometer operated with a capillary voltage of 4500 V in positive mode. Acquisitions were performed on the mass range *m/z* 200–1850. Calibration was performed using the singly charged ions produced by a solution of Tune mix (G1969-85000, Agilent, Clara, CA, USA). Compounds were separated on a XBridge Peptide BEH C18 column (300Å, 3.5 µm, 2.1 mm × 250 mm) column. The gradient was generated at a flow rate of 250 µL/min, at 60 °C by mixing two mobile phases. Phase A consisted of 0.1% formic acid (FA) in water and phase B of 0.08% FA in ACN. Phase B was increased from 5 to 85% in 45 min. Data analysis was performed using Compass DataAnalysis 4.3 (Bruker Daltonics, Billerica, MA, USA).

## 4. Conclusions

In this work, our aim was to introduce another chemical asset on our lead molecule plasmodione that can have a positive impact on the PK properties and can act as a shield for any detrimental effect coming from the complex host biological environment, in particular for long-term use of drug treatments. Thereby, from a Diels–Alder reaction we described the synthesis of racemic non-quinonoid Ψ-plasmodione regioisomers, 4a-Ψ-plasmodione **1** and 8a-Ψ-plasmodione **2**. The real breakthrough to control the regiochemical course of the reaction was achieved by the introduction of a boronic acid on the quinone as the auxiliary group to allow the full control of the cycloaddition regioselectivity. The next challenge will be to decorate the Ψ-plasmodiones with substituents that will influence the oxidation of angular methyl, but above all the cleavage of the C-C bond allowing aromatization and release of plasmodione.

## Data Availability

The original contributions presented in the study are included in the article/Appendix A, further inquiries can be directed to the corresponding authors.

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
