# Peer review of "Regioselective Synthesis of Potential Non-Quinonoid Prodrugs of Plasmodione: Antiparasitic Properties Against Two Hemoglobin-Feeding Parasites and Drug Metabolism Studies"

_molecules, 2024, doi:10.3390/molecules29225268_

Round 1
Reviewer 1 Report
Comments and Suggestions for Authors
This is an important contribution in the area of development of new compounds with anti parasitic effect. I recommend its acceptance. I also suggest to incoporate at the discussion comments on the possible advantage to incorporate Borum in the molecule in view of recent results showing the the presnce of thie element signifiacntly increase terapeutic activity agains some parasittic protozoa.
Reviewer 2 Report
Comments and Suggestions for Authors
The manuscript entitled “Regioselective synthesis of potential non-quinonoid prodrugs of plasmodione: antiparasitic properties against two hemoglobin-feeding parasites and drug metabolism studies” has been documented convincingly with the following observations.
1. Reactivity of Sulfinylquinones: A major limitation is the insufficient reactivity of sulfinylquinones towards Diels-Alder reaction. This issue stems from steric hindrance, which impedes their effectiveness in the synthetic process.
- Yield of the Synthesized Compounds: Specifically, 4a- and 8a-Ψ-plasmodione regioisomers were produced with overall yields of only 10.5% and 4.1%, respectively which is moderate enough but has scope for further optimization.
- Complexity of Biological Environment: The study noted that the biological environment can greatly influence the pharmacokinetics (PK) of the synthesized compounds. Complex interactions within a biological system may affect the performance of prodrugs, presenting a challenge for their long-term use in treatment.
- Limited Scope of Antiparasitic Testing: The Antiparasitic properties were mainly evaluated against Plasmodium falciparum and Schistosoma mansoni. This narrow focus may not fully capture the efficacy of compounds against a wider range of parasites, which is crucial for assessing their potential as therapeutic agents.
Comments to the Author:
1. Apart from the above notes, concern is on the supporting information portion. Some of the NMRs (Ex: 1HNMR of “27”) are questioning the quality of product and in turn will eventually influence the final efficiency.
2. Few 13C-NMRs need to be rerun with more number of scans (Ex: 13C-NMR of “27”).
3. Also suggested to authors to add one or more refence for the text presented in lines numbered 50 to 55 in the introduction.
I suggest that the authors relook into the comments and act accordingly.
Finally, I strongly recommend the manuscript to accept with minor revisions.
Reviewer 3 Report
Comments and Suggestions for Authors
In this manuscript, the authors synthesised of Ψ-plasmodione regioisomers as prodrugs of the antimalarial plasmodione. The presence of a chiral center adds a further dimension to their study on enantioselectivity and regioselectivity of D-A cycloaddition. The hypothesis of the research works is good, and the authors achieved promising results e.g., very fast cycloaddition time: ~2 h were obtained with boronic acid based quinones 25 and 27 in the presence of 1-methoxy-1,3-butadiene, to afford the 4a- and 8a-Ψ-plasmodione regioisomers 1 and 2 with an overall yield of 10.5% and 4.1%, respectively. I find this study important and acceptable for Molecules with minor revision, noted.
However I have some comments for the authors which I believe could add quality to the present manuscript at the revision stage.
IC50 values are normal and in the level of micromolar. But why the SD are so high? the high SD indicates lack of precision. Can the authors comment on it? If needed new experiment can be considered.
Metabolic studies are acceptable.
New sulfinylquinones were synthesized but were found to be ineffective in undergoing cycloaddition with different dienes under a wide range of conditions (thermal, Lewis acid). The thermal cycloaddition of the sulfone containing PAHs are quite known (. Am. Chem. Soc. 1978, 100, 5, 1597–1599). To my surprise that did not work here. Could the authors please explain or suggest a possible explanation to it to merit this fact?
The literature span is too narrow. For stability/occupancy of radicals on chiral hydrocarbons, the following paper must be cited: Chemical Communications, 2019, 55, 43, 6022-6025.
Chiral compounds' radical polymerization and the effect of chirality is known and should to be cited from recent literature: The Journal of Physical Chemistry C 2020, 124 (38), 20974-20980.
Round 2
Reviewer 3 Report
Comments and Suggestions for Authors
Based on the recent version, this manuscript can be accepted.